# An Underwater Visual Navigation Method Based on Multiple ArUco Markers

Zhizun Xu [1,*], Maryam Haroutunian [2], Alan J. Murphy [2], Jeff Neasham [2] and Rose Norman [2]

1  Maritime College, Guangdong Ocean University, Zhanjiang 524000, China
2  School of Engineering, Newcastle University, Newcastle upon Tyne NE1 7RU, UK;
   maryam.haroutunian@newcastle.ac.uk (M.H.); a.j.murphy@newcastle.ac.uk (A.J.M.);
   jeff.neasham@newcastle.ac.uk (J.N.); rose.norman@newcastle.ac.uk (R.N.)
*  Correspondence: zhizun@gdou.edu.cn

**Abstract:** Underwater navigation presents crucial issues because of the rapid attenuation of electronic magnetic waves. The conventional underwater navigation methods are achieved by acoustic equipment, such as the ultra-short-baseline localisation systems and Doppler velocity logs, etc. However, they suffer from low fresh rate, low bandwidth, environmental disturbance and high cost. In the paper, a novel underwater visual navigation is investigated based on the multiple ArUco markers. Unlike other underwater navigation approaches based on the artificial markers, the noise model of the pose estimation of a single marker and an optimal algorithm of the multiple markers are developed to increase the precision of the method. The experimental tests are conducted in the towing tank. The results show that the proposed method is able to localise the underwater vehicle accurately.

**Keywords:** underwater navigation; artificial fiducial marker; visual localisation

## 1. Introduction

The oceans cover 71% of the earth. Unmanned Underwater Vehicles (UUVs) are widely implemented to explore the oceans. The reliable underwater localisation systems are essential to increase the efficiency of the UUVs in underwater missions. In contrast to the localisation and navigation methods applied by land robotics, underwater navigation is a challenging problem, mainly because the electromagnetic wave attenuates rapidly in the water. It indicates that the methods of localisation or communications depending on the electromagnetic wave are unsuitable in underwater environments. Therefore, conventional methods of positioning of UUVs are achieved by acoustic beacons, such as Ultra-Short-Baseline Localisation (USBL) or Long Baseline Localisation (LBL). However, these devices require additional infrastructure, which means the cost is fairly high [1]. In addition to that, the acoustic equipment, including the Doppler Velocity Log (DVL), suffers from a low refresh rate, low bandwidth and noises in underwater environments.

There are many reports on solving underwater navigation issues using visual techniques [2–5]. Researchers fused the visual information of the seabed with other data collected by inertial sensors to predict the positions and attitudes of the UUVs [6,7]. However, the limitations such as insufficient illumination, scattering, and refraction, prevent underwater visual navigation methods from being implemented.

In this paper, a novel underwater localisation method based on the multiple ArUco markers has been investigated. The ArUco markers, reported by Jurado, can be detected automatically, and their positions relative to the camera are obtained as well [8]. While the ArUco markers were intended to serve for the augmented reality applications originally, it is widely utilised to locate the mobile robots [9,10]. In the proposed method, multiple markers with known positions are set out on the ground of a towing tank. After calibration of the camera in water, the positions and attitudes of the camera relative to the markers can be obtained by inverting the transformation matrices. Subsequently, an optimal algorithm

will be used to update the position of the camera in the global coordinate system. Since the camera is fitted on the UUV, the positions of the UUVs are obtained by multiplying the geometric transformation matrices. The practical experiments have been conducted to prove that the proposed method is effective while being low-cost. The method is expected to be applied in underwater pipeline tracking or areas that are occupied by artificial structures. After placing the markers on the structures, the approach can estimate the positions and altitudes of UUVs precisely in the area.

The paper is structured as follows. The related work is outlined in Section 2. The introduction of ArUco markers and the testing platform are presented in Sections 3 and 4. Subsequently, the brief theory and the process of the underwater camera calibration are shown in Section 5. In Section 6, the methodology is described in detail, and the layout of the markers is shown in Section 7. The experimental results and discussion are presented in Section 8. Section 9 concludes the approach and discusses the future work.

## 2. Related Work

The visual navigation methods have been widely used by mobile robots and drones. Babinec reported a localisation method for mobile robots with the use of the ArUco markers deployed to the environment [9]. The results showed that this system was reliably employed in the visual localisation of mobile robots. Xing developed a multi-sensor fusion indoor localisation system based on the ArUco markers for mobile robotics. The sensors include: markers, optical flow, ultrasonic and the inertial sensor. The results showed that the proposed method has satisfactory performances [11]. In [10], the extended Kalman filter algorithm is utilised to fuse odometer information and the information from detection ArUco markers. Meng applied the ArUco markers to provide localisation services for indoor IoT (Internet of Things) applications [12].

The underwater visual navigation methods are also reported by many researchers, while the subsea environments are harsh for optical sensors. A real-time Monocular Visual Odometry for underwater vehicles was developed by Ferrera. In the method, the Optical Flow algorithm was utilised to track feature points [13]. In [7], Leutenegger improved the Open Keyframe-based Visual Inertial SLAM(OKVIS) to locate the underwater vehicles by using an underwater profiling sonar. The information of a stereo camera, a profiling sonar, an IMU and a pressure sensor were combined to construct a cost function. The movement and rotation of the vehicle can be estimated by minimising the cost function. In [14], Rahman utilised the image enhance technique and a loop-closure technique to improve the performance of underwater visual navigation. However, these methods generally suffered from accumulated errors.

The first example of using the artificial fiducial markers to work as a way of underwater navigation is Morales, who investigated the advantages of the markers to provide visual cues, such as an artificial horizon or navigation arrows, to assist underwater operations [15]. Later, Jasiobedzki utilised the markers to recognise and track the position of underwater vehicles in poor visibility conditions [16].

Dos made a comparison between the performances of three different artificial fiducial markers: ARToolKit, AprilTags, and ArUco [17]. He found that all of them presented slow performances in underwater environments. Therefore, many researchers started using computer vision techniques to improve the quality of images captured in underwater environments. In [18], Cejka improved the performance of AcUco in underwater environments by adjusting the threshold in the detection step. He reduced the noised contours and weakened the background, compared with the original method. The experimental results showed that the new method performed better than original one. In [19], Zuzi used the three dehazing techniques to enhance the underwater images where there were artificial markers. The experimental results showed that SP (Screened Poisson Equation for image contrast enhancement) outperformed the other two enhancement algorithms, BCP (Bright Channel Prior) and ACE (Automatic Color Enhancement), while all of them presented that the marker detection was completed in a shorter time. Agarwal increased

the contrast of images to improve the performances of the visual positioning based on artificial fiducial markers [20]. Da discussed the error distributions of visual markers in underwater environments in his master dissertation [21]. After the detection of the ArUco markers is characterised, the detection capacity on the part of the ArUco library is weak from a certain pitch angle of inclination of the AUV. Hence, he developed a method to improve the marker detection and extracted more information from the markers. Ren used a Kernel Correlation Filter (KCF) acceleration strategy for short sensing time [22]. A pool experiment was conducted to show the advantages of the method.

Meanwhile, the information of artificial fiducial markers is applied to be fused with the data collected by other navigation sensors. Barbera used a particle filter method to fuse the sensor information from the ArUco markers, the imaging sonar and the ping sonar. The underwater vehicles were located by using a Sequential Monte Carlo method initialised from GPS location acquisition on the surface [23]. The results showed that the method was a reliable solution for underwater navigation. Chavez implemented the ArUco markers to boost the navigation performance [24]. In his work, a navigation system, consisting of an EKF, a DVL and an inertial navigation system, is extended by visual odometry using artificial markers. The systems were evaluated in two intensive field tests. The results presented that the implementation of the markers did increase navigation performance. In addition to that, Zhang used the ArUco marker to estimate the position of the manipulator at a high rate of speed in underwater environments [25]. Experiments are carried out with the 7-function underwater hydraulic manipulator. The results showed that the ArUco marker was able to predict the accurate position for the controller of the underwater manipulator.

Unlike the above-mentioned works, in the paper, a novel underwater visual navigation method is investigated based on the multiple of ArUco markers. In the proposed method, a noise model of the estimation from a single marker is built. An optimal algorithm is derived to fuse the information extracted from the multiple markers to improve the accuracy of the method.

## 3. Introduction of ArUco Markers

There are three well-known artificial fiducial marker systems: ARToolKit [26], April-Tags [27], and ArUco [8]. Compared with others, the ArUco algorithm is able to detect and track the markers quickly and reliably. As shown in Figure 1, an adaptive threshold selection algorithm is implemented by computing the average value of the surrounding pixels of the specific pixel. In this way, all contours in the image are expected to be found, and non-square contours are filtered out. Followed by the projection operation using recognised contours, the code extraction, marker identification and error correction are processed. Eventually, the unique ID and the pose of the marker relative to the camera are estimated.

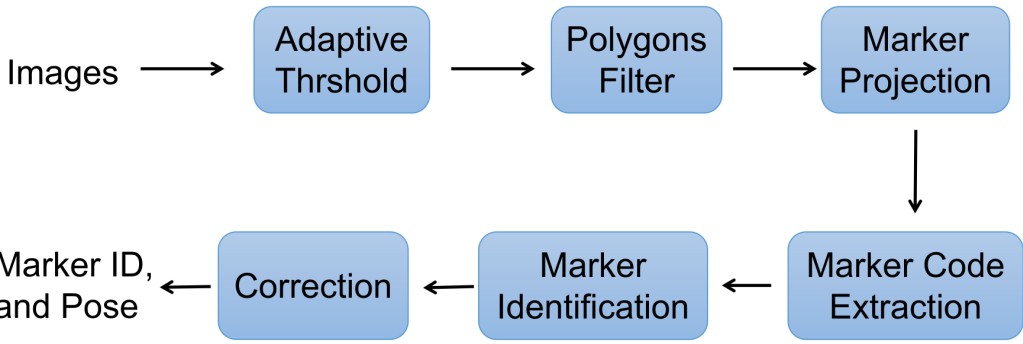

**Figure 1.** The ArUco Algorithm.

## 4. Testing Platform

The modified Videoray Pro 3 ROV was used to collect the data in the towing tank [28]. A stereo camera (Intel T265 Tracking camera) and an onboard computer (Latte Panda) are installed inside the watertight tube, which is mounted on the bottom of the Videoray Pro 3, as shown in Figure 2. During the collection process, the vehicle was operated remotely by a controller to follow specific paths. After tests, the collected data can be transferred to the terminal via WiFi.

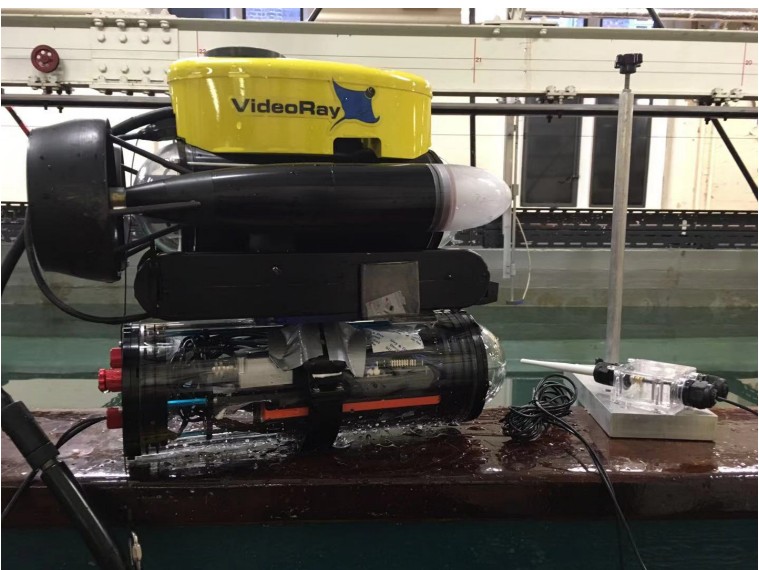

**Figure 2.** VideoRay Pro 3 with tube.

## 5. Underwater Camera Calibration

The underwater camera calibration is essential for visual navigation systems. It not only can handle the distortion issue induced by the lenses but is also used to reduce the refraction effect. The camera calibration problem is solved by minimising the cost function [29]:

$$E = \frac{1}{2} \sum_{i=1}^{m} [H(\mathbf{\Theta}, g_i) - p_i]^2, \tag{1}$$

where a grid based on a chessboard consists of *m* points of $g_i$ coordinates with corresponding values $p_i$. The term *H* is a function mapping grid points into the image plane. The $\mathbf{\Theta}$ involves intrinsic, distortion, and extrinsic parameters.

The Levenberg–Marquardt approach can be applied to minimise the cost Function (1), solving for $\mathbf{\Theta}$. In order to minimise the cost function effectively, the initial estimates of camera parameters are processed by Direct Linear Translation (DLT) [30].

In practice, the camera calibration is processed by OpenCV tools, which are developed based on [31]. In Figure 3a, a classic black-white chessboard was placed on the bottom of the towing tank, and the vehicle was operated remotely to obtain images of the chessboard from different views. The process of the calibration through the chessboard pattern is shown in Figure 3b. The comparison is illustrated in Figure 3c,d. They display the original image and the calibrated image, respectively.

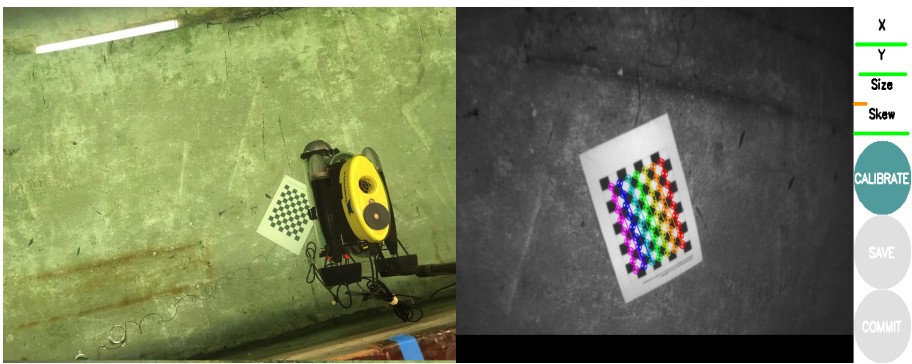

(**a**) Chessboard and Videoray Pro 3        (**b**) Calibration Samples

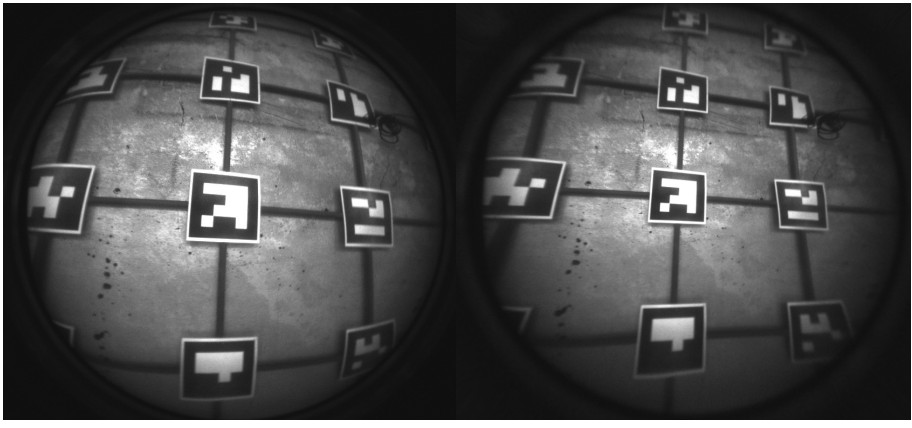

(**c**) Original Image        (**d**) Calibrated Image

**Figure 3.** (**a**) The chessboard on the ground and the vehicle are operated to capture the images. (**b**) The calibration is processed based on the captured patterns. (**c**) The original image. (**d**) The same image after being calibrated.

## 6. Methodology

In the proposed method, the positions of the camera relative to different single markers are derived by using transformation matrices. Because the positions of the markers in the global coordinate system are known beforehand, the position of the camera in the global coordinate system is expected to be estimated. However, the positions measured by different markers do not indicate the real value due to the noise of underwater imaging. Hence, the noise model for the estimation by a single marker is constructed, and an optimal algorithm is designed to combine the predicted poses from the multiple markers to increase the accuracy of the localisation, as shown in Figure 4.

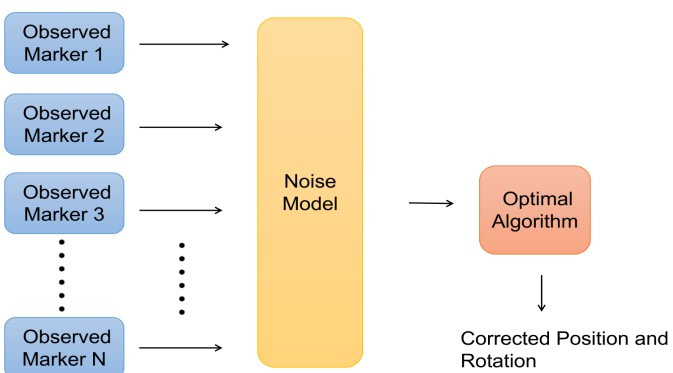

**Figure 4.** The flow chart of the method.

### 6.1. Position Estimated by the Single Marker

The ArUco libraries from OpenCV were implemented in this navigation method. By calling the estimatedPoseSingleMarkers function in the libraries, the rotation vector and translational vector of the marker relative to the camera were expected to be obtained. With the Rodrigues' formula [32], the rotation vector was converted to the associated rotation matrix. Hence, the transformation matrix $^{camera}T_{marker}$ was derived. However, $^{marker}T_{camera}$ was needed in order to acquire the position and the attitude of the camera in a marker's coordinate system.

In the free-noise phenomenon, the $^{marker}T_{camera}$ can be derived by,

$$^{marker}T_{camera} = {}^{camera}T_{marker}^{-1}.$$ (2)

Because

$$^{camera}T_{marker} = \begin{bmatrix} R_{mc} & t_{mc} \\ 0 & 1 \end{bmatrix}$$ (3)

Then,

$$^{marker}T_{camera} = \begin{bmatrix} R_{mc}^T & -R_{mc}^T t_{mc} \\ 0 & 1 \end{bmatrix}.$$ (4)

where $u_m = -R_{mc}^T t_{mc}$ is the translational vector of the camera relative to the marker. The term $R_{mc}^T$ is the rotation matrix of the camera relative to the marker.

Before investigation about the white noise situation, the Lie algebra for the rotation matrix should be discussed. The $R_{mc}$ has nine individual entries, and eight degrees of freedom, because it is in the Special Orthogonal Group, i.e., $R \in SO(3)$ and $R^T R = I$. The Lie algebra $\phi$, a 3-element vector, can be used to present the rotation matrix $R_{mc}$ [33]. Since the mapping from the rotation matrix $R$ to Lie algebra $\phi$ complies with the Rodrigues' formula [32], the Lie algebra $\phi$ actually equals the rotation vector.

The exponential mapping for $R$ and $\phi$ is,

$$R = exp(\phi^\wedge).$$ (5)

The $\phi^\wedge$ denotes the skew-symmetric matrix of the vector $\phi$. The Lie algebra $\overline{\phi}$ of the rotation matrix $R^T$ is,

$$\overline{\phi} = (\ln(R^T))^\vee = (\ln exp(\phi^\wedge)^{-1})^\vee = -\phi$$ (6)

Hence, the $^{camera}T_{marker}$ can be presented by $\{\phi_{mc}, t_m c\}$, the $^{marker}T_{camera}$ can be presented by $\{-\phi_{mc}, exp(-\phi_{mc})t_{mc}\}$.

As mentioned before, the noise caused by underwater imaging has been introduced into the translational vector $t_{mc}$, and rotation matrix $\phi_{mc}$. According to [30,34],

$$\{\hat{\phi}_{mc}, \hat{t}_{mc}\} \sim N(\{\phi_{mc}, t_{mc}\}, \Sigma_{mc})$$ (7)

and,

$$\Sigma_{mc} = \begin{bmatrix} Q_1 & 0 \\ 0 & Q_2 \end{bmatrix}$$ (8)

That means the $\{\phi_{mc}, t_{mc}\}$ obeys the Gaussian distribution. The $\Sigma_{mc}$ is a diagonal positive semidefinite matrix. The $Q_1$ and $Q_2$ are $3 \times 3$ matrices. The diagonal entries are constructed by the squared exponential covariance function [35], which is,

$$k_i(r_i) = exp(-\frac{r_i^2}{2l_i^2})$$ (9)

where the $k_i(r_i)$ is the kernel function associated with the covariance function. The $r_i$ is the distance or radius on the xyz axis estimated by a single ArUco marker. The $\Sigma = diag(k_i^{-1} - 1)$. In this case, the $|r_i| \to 0$, the $\Sigma_i \to 0$. The $l_i$ is a hyperparameter defining the characteristic

length-scale. In the paper, it is selected as 1 for estimated coordinates and 0.5 for the estimated rotation vectors. That means the smaller estimated value is more reliable.

Hence, according to [36], the probabilistic model of $\{\boldsymbol{\phi}_{cm}, \boldsymbol{t}_{cm}\}$ can be expressed as,

$$\{\hat{\boldsymbol{\phi}}_{cm}, \hat{\boldsymbol{t}}_{cm}\} \sim N(\{-\boldsymbol{\phi}_{mc}, exp(-\boldsymbol{\phi}_{mc}^{\wedge})\boldsymbol{t}_{mc}\}, \boldsymbol{\Sigma}_{cm}) \tag{10}$$

where the $\hat{\boldsymbol{\phi}}_{cm}$ and $\hat{\boldsymbol{t}}$ are random variables and,

$$\boldsymbol{\Sigma}_{cm} = \begin{bmatrix} Q_1 & 0 \\ 0 & M \end{bmatrix} \tag{11}$$

However, the derivation of $\boldsymbol{M}$ is a little trivial. The $exp(-\hat{\boldsymbol{\phi}}_{mc}^{\wedge})$ is a nonlinear transform with respect to random variable $\hat{\boldsymbol{\phi}}_{mc}$, and $exp(-\hat{\boldsymbol{\phi}}_{mc})\hat{\boldsymbol{t}}_{mc}$ is a production of dual Gaussian distributions. According to [36], $\boldsymbol{M}$ can be,

$$\boldsymbol{M} = [-exp(-\boldsymbol{\phi}_{cm}^{\wedge})\boldsymbol{t}_{cm})^{\wedge} \, exp(-\boldsymbol{\phi}_{cm}^{\wedge})]Q_2[-exp(-\boldsymbol{\phi}_{cm}^{\wedge})\boldsymbol{t}_{cm})^{\wedge} \, exp(-\boldsymbol{\phi}_{cm}^{\wedge})]^T \tag{12}$$

Now the derivation of position estimation by a single ArUco marker in the white noise situation is complete. In the next part, an optimal algorithm is described, which is used to increase the accuracy of the method based on the multiple markers.

### 6.2. The Optimal Algorithm Based on Multiple Markers

The ArUco algorithm can detect multiple markers and estimate the relative poses simultaneously. Assuming N markers observed, there are N numbers of translational and rotation vectors, i.e., $\{\boldsymbol{\phi}_i, \boldsymbol{t}_i\}$ and $0 < i \leq N$. An optimal algorithm was designed based on multiple markers to upgrade the position estimated from a single marker. The transform function $Tf^{-1}$ maps the coordinate from the reference of a single marker into the global reference. Because of the layout of the markers, the rotation vectors estimated from different markers are in the same reference. The detailed interpretation is presented in Section 7.

In the optimal algorithm, the cost function was built by employing the Mahalanobis distance. Subsequently, the $\tilde{\boldsymbol{\phi}}$ and $\tilde{\boldsymbol{t}}$ need to be derived to minimise the Mahalanobis distance. There are other methods to achieve that, for example: the well-known RANSAC (Random Sample Consensus), and the weighted average method. However, they may both cause large errors in estimated poses. The marker closer to the camera can provide more accurate pose estimation in its coordinate system. In some cases, the RANSAC and weighted average methods may fail in finding the optimal pose, for instance, when the most similar observed values are estimated by markers at large distances. In the proposed method, once constructing the covariance function properly, the algorithm of minimisation of the Mahalanobis distance is expected to obtain the optimal pose successfully.

The Mahalanobis distance can be expressed as,

$$l = \sum_{i=1}^{N}(\{\boldsymbol{\phi}_i, Tf^{-1}(Id) + \boldsymbol{t}_i)\} - \{\tilde{\boldsymbol{\phi}}, \tilde{\boldsymbol{t}}_g\})^T \boldsymbol{\Sigma}_i^{-1}(\{\boldsymbol{\phi}_i, Tf^{-1}(Id) + \boldsymbol{t}_i)\} - \{\tilde{\boldsymbol{\phi}}, \tilde{\boldsymbol{t}}_g\}) \tag{13}$$

where the $\tilde{\boldsymbol{\phi}}$ is the optimal rotation vector, and $\tilde{\boldsymbol{t}}_g$ is the optimal translational vector in the global reference. The $\boldsymbol{\Sigma}_i$ is identical to $\boldsymbol{\Sigma}_{cm}$,

To minimise the Mahalanobis distance, let the differential of $l$ be equal 0 with respect to $\{\tilde{\boldsymbol{\phi}}, \tilde{\boldsymbol{t}}_g\}$,

$$\frac{\partial l}{\partial\{\tilde{\boldsymbol{\phi}}, \tilde{\boldsymbol{t}}_g\}} = 0. \tag{14}$$

The $\{\tilde{\boldsymbol{\phi}}, \tilde{\boldsymbol{t}}_g\}$ can be derived as,

$$\{\tilde{\boldsymbol{\phi}}, \tilde{\boldsymbol{t}}_g\} = (\sum_{i=1}^{N} \boldsymbol{\Sigma}_i)^{-1}(\sum_{i=1}^{N} \boldsymbol{\Sigma}_i^{-1}\{\boldsymbol{\phi}_i, \boldsymbol{t}_i\}). \tag{15}$$

Using the optimal algorithm, the optimal poses of the camera based on the multiple markers are derived. In the following section, the layout of the multiple markers will be introduced.

## 7. Layout of the Multiple Markers

In the towing tank, the multiple markers are set out on the ground. The layout of the markers is shown in Figure 5, and the reference of the markers is plotted in Figure 6.

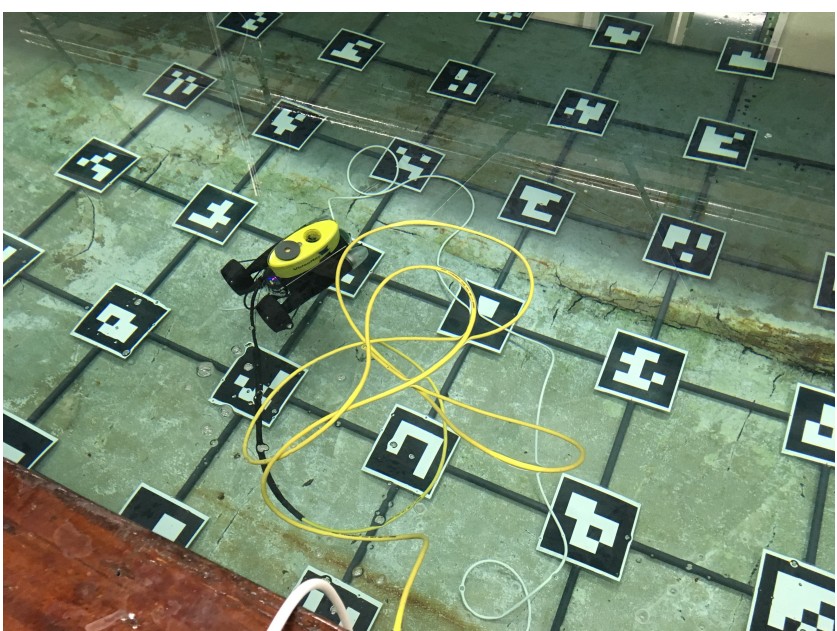

**Figure 5.** Markers in the towing tank.

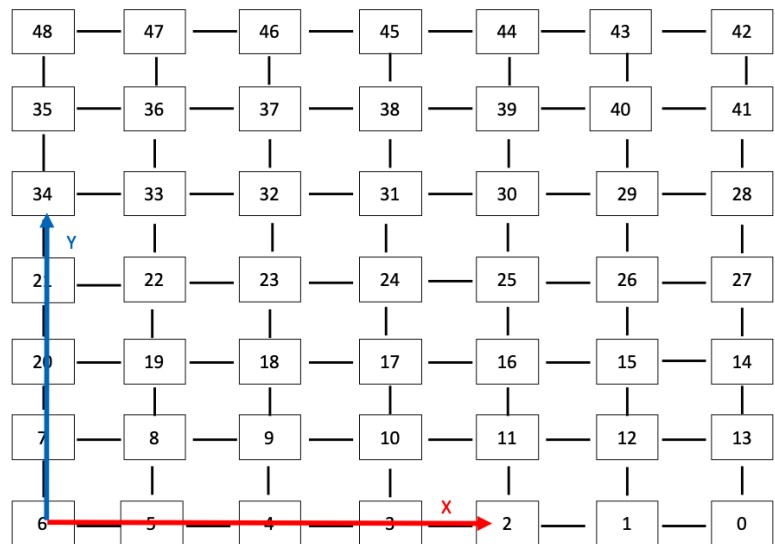

**Figure 6.** Reference of the markers.

According to Figure 6, the origin of the global coordinate system is located at the Id.6 marker. After obtaining the relative pose to a specific marker, the transfer function ($Tf$) is derived as,

$$Id = Tf(r, c) = 6r + (-1)^r(6 - c) \tag{16}$$

where $Id$ is the unique ID of the marker. Symbols $r$ and $c$ are the $r$th row and $c$th column, respectively.

The inverse function of $Tf$ can be used to obtain $r$ and $c$, $Id \rightarrow (r, c)$ with $Tf^{-1}$. The inverse function $Tf^{-1}$ can be derived by the following two equations,

$$r = [\frac{Id}{6}];$$ (17)

$$c = (-1)^{r+1}(Id - 6r) + 6.$$ (18)

where $[\ ]$ indicates the integer division operator. Using these equations, the $(c, r)$ is calculated. Because the maximum distance of a single marker is about half a metre, the distance between markers on the grid is 0.5 m, the position of the marker in the global coordinate system is known by multiplying the constant scalar 0.5 by $c$ and $r$ as follows,

$$x_m = 0.5c;$$ (19)

$$y_m = 0.5r.$$ (20)

The camera pose relative to the marker ($u_m$), with the position of the marker in the global reference ($x_m, y_m, 0$), can be transferred to $u_g$ in the global coordinate system. Because all markers are parallel, the rotation matrix remains the same. Hence, the derivation of the camera pose in the global reference frame is completed.

## 8. Experimental Results and Discussion

As mentioned before, the experiments were conducted in the towing tank. The vehicle introduced in Section 4 was controlled by an operator remotely. The proposed underwater visual navigation was tested five times: two times with lawnmower patterns, three times with random closed shapes, and one time with a random shape. The navigation system provides the 3D trajectory in absolute scale, as shown in Figure 7.

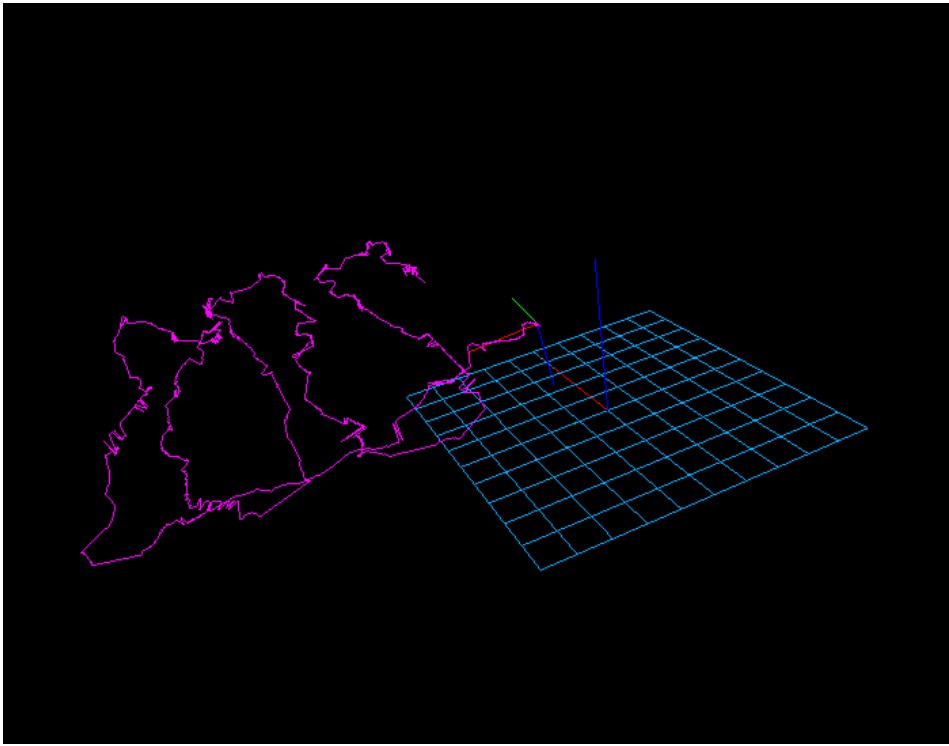

**Figure 7.** The 3D trajectory estimated by the method.

Since the vehicle was mainly controlled to move in the horizontal plane, the 3D trajectories were converted into the 2D plots, as shown in Figures 8–13.

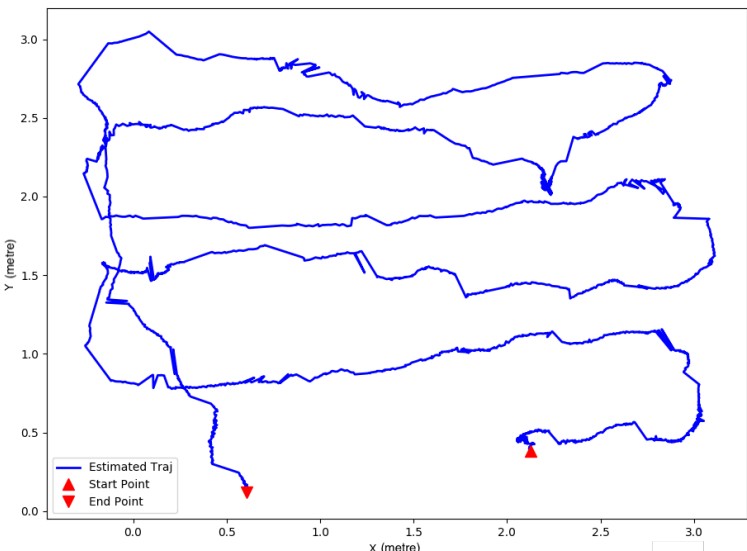

**Figure 8.** Lawnmower pattern—1st.

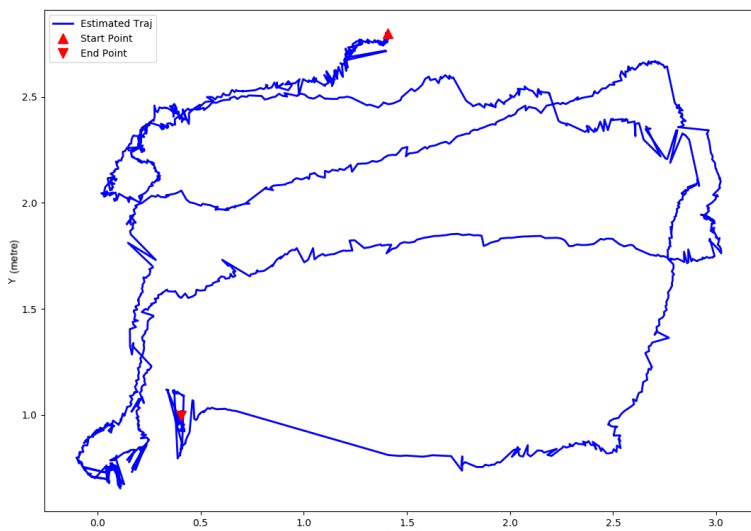

**Figure 9.** Lawnmower pattern—2nd.

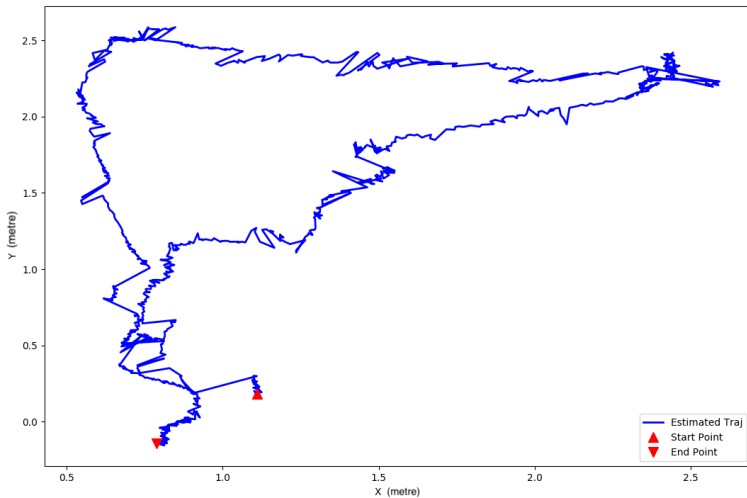

**Figure 10.** Closed trajectory—1st.

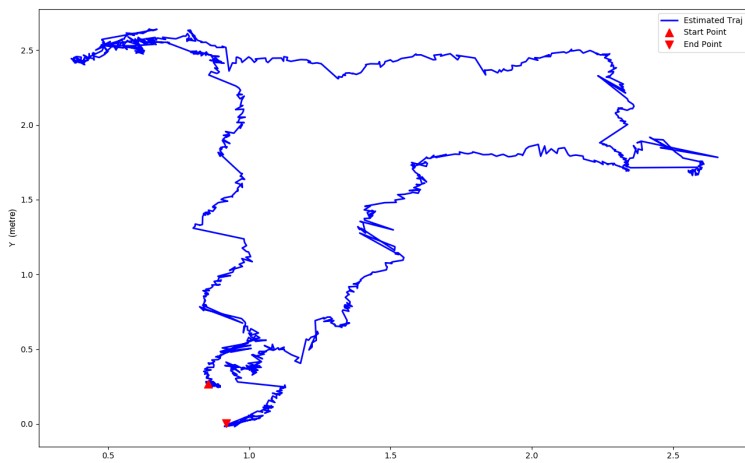

**Figure 11.** Closed trajectory—2nd.

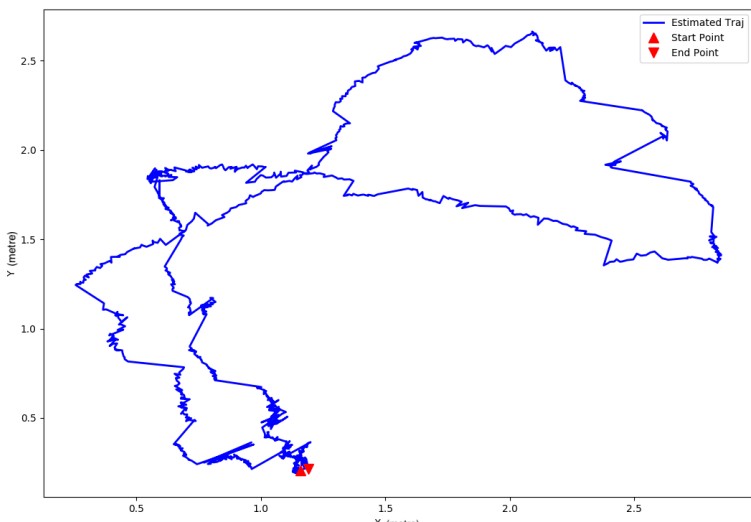

**Figure 12.** Closed trajectory—3rd.

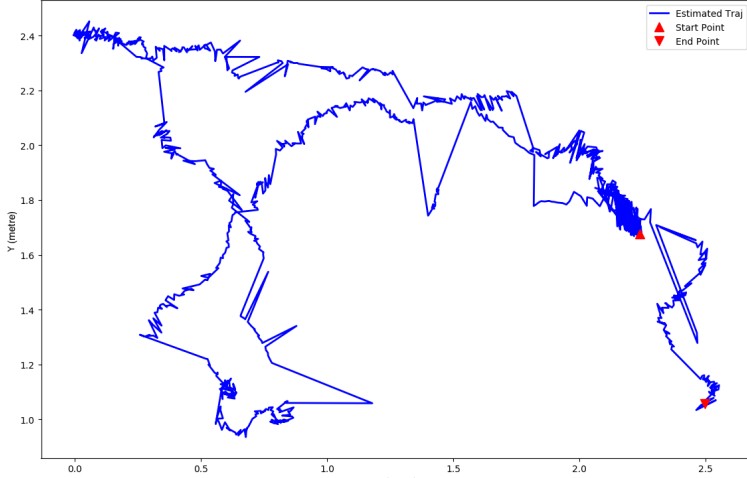

**Figure 13.** Random trajectory.

In Figures 8 and 9, the vehicle was operated to travel along an approximated lawn-mower path. The predicted trajectories in the figures mostly follow lawnmower patterns. The length of the vehicle is about 0.2 m. In Figure 8, the vehicle travelled along the

lawnmower pattern and returned on the same side. In Figure 9, after completing the lawnmower-pattern journey, the vehicle docked on the opposite side. The results show the proposed method presents reasonable estimated trajectories.

In Figures 10–12, the vehicle was controlled to complete a closed-loop path. These closed trajectories are generated arbitrarily by the operator. In Figure 12, the estimated trajectory is closed exactly at the end. In Figures 10 and 11, there are small deviations between the start point and the end point in the estimated trajectories. As mentioned before, the length of the vehicle is over 0.2 m. The offsets may be caused by the operation error controlling the ROV or by hydrodynamic disturbances (waves, current). After that, the vehicle was controlled to complete a random trajectory, and the result is shown in Figure 13.

Unlike odometry methods, which suffer from unbounded cumulative error, the navigation method based on the ArUco markers is not dependent on the previous states. Hence, the error of the algorithm is bounded. However, in Figure 13, there are discontinuities in the estimated trajectory. These are caused by mistakes in the marker detection algorithm. The detection algorithm may recognise the marker incorrectly when the marker is far from the camera or is only partially visible. Because of this, the trajectories generated by the marker navigation system are not consistent.

For the quantitative evaluation, the positions of the markers in images can indicate the real position of the camera at that time. Three examples are selected randomly, shown in Table 1, where the markers' positions and estimated positions are listed. The markers' positions are known beforehand, and the camera positions are estimated by the proposed method. Through the comparison in Table 1, it is clear that the estimated position is close to the position of the corresponding marker. The deviations between the marker positions and estimated positions are less than 0.5 m, and most of them are about 0.2 m. Since the camera is not exactly on the centre top of the markers, the deviations are acceptable. According to Table 1, the localisation error of the proposed approach is bounded, and the accuracy of the method is guaranteed.

**Table 1.** Underwater ArUco method quantitative test (unit: metre).

| Images | Marker Position | Estimated Position |
|---|---|---|
|  | (2.5, 2.5) | (2.73, 2.97) |
|  | (1.5, 1.5) | (1.61, 1.67) |
|  | (2.0, 2.5) | (1.86, 2.30) |

## 9. Conclusions

In the paper, the detection algorithm of the ArUco markers is introduced first. Then, the testing platform and underwater camera calibration are presented as well. Based on the above, a novel underwater visual navigation method using the multiple ArUco markers is investigated. In the method, the noise model of the estimation by a single marker and an optimal algorithm for multiple markers are derived. The experimental results showed that the proposed method can estimate the positions and altitudes of UUVs precisely. The main contributions of the paper are listed as follows.

- The multiple ArUco markers are utilised to achieve the accurate localisation of UUVs in certain underwater areas;
- The probabilistic model of the position estimated by the ArUco is constructed by introducing the Lie algebra.

The advantage of the marker navigation over odometry is that the position error is bounded. In the underwater environments, where the artificial structures are laid out, the markers can be placed in the structures. In this case, the accurate positioning can be implemented in the area by using the proposed method. However, there is some chattering in the estimated trajectories. Some parts of estimated trajectories are not consistent. In future work, a smoothing algorithm is expected to be utilised to solve the issue.

**Author Contributions:** Conceptualization, Z.X. and A.J.M.; methodology, Z.X.; software, Z.X.; validation, Z.X.; formal analysis, Z.X.; investigation, Z.X.; resources, Z.X., J.N., A.J.M., M.H. and R.N.; data curation, J.N. and R.N.; writing—original draft preparation, Z.X.; writing—review and editing, Z.X.; visualization, Z.X.; supervision, R.N.; project administration, R.N.; funding acquisition, R.N. All authors have read and agreed to the published version of the manuscript.

**Funding:** This research received no external funding.

**Institutional Review Board Statement:** Not applicable.

**Informed Consent Statement:** Not applicable.

**Data Availability Statement:** Not applicable.

**Acknowledgments:** The authors would like to acknowledge the generous support of John and Vivien Prime in funding aspects of this work.

**Conflicts of Interest:** The authors declare no conflict of interest.

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
