# Peer review of "An Underwater Visual Navigation Method Based on Multiple ArUco Markers"

_jmse, doi:10.3390/jmse9121432_

Round 1

Reviewer 1 Report

  1. In Fig. 9-14, the estimated trajectories are shown but it is not clear from the diagrams whether it is estimating the actual trajectories precisely. No reference plots are there for actual trajectories.
  2. The experiments are performed in known environment without any obstruction but in real case, environment will be totally unknown and there will be various kinds of obstruction due to marine environments. Applicability of the proposed technique in real scenario is not clear.
  3. No clear indication about the maximum distance from which this navigation can be performed.
  4. It is stated that it is better as compared to many existing works but no comparison is done.
  5. There are many grammatical and typo errors (abstracts, introduction, etc sections)
  6. References are not uniformly listed.

Reviewer 2 Report

Very interesting article, the method probably still needs to collect experiments from different reservoirs under different disturbances. In future work I suggest working on smoothing the solution trajectory.

Author Response

Thanks for your comments. For the underwater localisation system, the disturbance are caused by the scattering and the uncertain illumination. The authors will consider the issues in the future work.

Reviewer 3 Report

The findings in this paper are not new.

Author Response

In the method, the noise model of the estimation by a single marker and an optimal algorithm for multiple markers are derived. The experimental results showed the proposed method can estimate the positions and altitudes of UUVs precisely. The main contributions of the paper are listed as follows.

(1) The multiple ArUco markers are utilised to achieve the accurate localisation of UUVs in certain underwater area;

(2) The probabilistic model of the position estimated by the ArUco is constructed by introducing the Lie algebra.

The authors have emphasised the novelties in the conclusion section on Page 14, Line 295 to 312.

Reviewer 4 Report

Good job, good scientific article. However, it is worth making a few adjustments:

  • The section "Introduction" has too much obvious unnecessary content and should be redrafted.
  • Figure 7 adds nothing new and is redundant.
  • The "Conclusions" section seems to be overly brief and does not fully emphasize the valuable results obtained. It should be slightly supplemented and redrafted.

Round 2

Reviewer 1 Report

Revised version is fine. Only concern is that distance from which navigation can be done is only 0.5 m which is too small. I don't see any merit to go for navigation from such a short distance.

Author Response

Thanks for your comments. The maximum distance for the single marker is 0.5 metres. Because the optical camera suffers from the scattering and the low illumination in the underwater environments.  However, the paper mainly discussed about the multiple ArUco markers.  Unlike the acoustic beacons, the cost of the markers is extremely low. When many markers are set out in the certain area, the high accurate positioning can be achieved.

Reviewer 3 Report

Please use EKF localizing and particle filter. The path of the vehicle is very unsteady and therefore inefficient. Please have a look at the following book:

Author Response

The paper focuses on the probabilistic model construction of the estimation from the single marker and  investigating the multiple ArUco marker navigation system. The EKF and particle filter are widely used in robotic field, especially for sensors fusion problems, which have been mentioned in "related work" section. The accuracy and efficiency of the method have been discussed on Page 13 Line 284 to 293. The authors will use the smoothing algorithm or sliding window algorithm to solve the chattering issue in future, which is mentioned in conclusion section.

Round 3

Reviewer 3 Report

please have a look at https://docs.ufpr.br/~danielsantos/ProbabilisticRobotics.pdf

page 180, chapter 7.5.2. Your path-planning can be improved.

Author Response

Thanks for your comments. The authors have read the section which you recommended. It declared the EKF algorithm for robotics positioning. The EKF has been mentioned in "related works" section in the paper. The EKF mainly talks about the prediction based on the dynamic model, and the correction based on observation measurements. However, such pattern is unsuitable for the proposed method. The method is designed by using the  single optical camera. The performance of the method  has been evaluated and efficiency of the method has been proved as well on Page 13, Line 284 to 293.